# Evaluation of Single-Cropping under Reduced Water Supply in Strawberry Cultivation

Micol Marcellini [1], Luca Mazzoni [1], Davide Raffaelli [1], Valeria Pergolotti [1], Francesca Balducci [1], Franco Capocasa [1] and Bruno Mezzetti [1,2,*]

[1] Department of Agricultural, Food and Environmental Sciences, Università Politecnica delle Marche (UNIVPM), Via Brecce Bianche 10, 60131 Ancona, Italy; micol.marcellini@staff.univpm.it (M.M.); l.mazzoni@staff.univpm.it (L.M.); s1107943@pm.univpm.it (D.R.); v.pergolotti@pm.univpm.it (V.P.); francesca.balducci@staff.univpm.it (F.B.); f.capocasa@staff.univpm.it (F.C.)

[2] UNEA, Research Group on Food, Nutritional Biochemistry and Health, Universidad Europea del Atlántico, Isabel Torres, 21, 39011 Santander, Spain

* Correspondence: b.mezzetti@staff.univpm.it; Tel.: +39-0712204640

**Abstract:** Genotype, environment, and cultivation system strongly influence strawberry yield and quality. Specifically, the growth of strawberry plants is dependent on the water supply. Nevertheless, the abuse of water in agriculture is necessitating the choice of the lowest water-consumptive plants. The following study showed the performance of 'Romina', 'Sibilla', and 'Cristina' cultivars, grown in open-field conditions, and treated with three doses of water (W): 100% local standard regime, and 20% (W80) and 40% (W60) reductions. The average amount of water administered for W100, W80, and W60 was 1120 $m^3$ $ha^{-1}$, 891 $m^3$ $ha^{-1}$, and 666 $m^3$ $ha^{-1}$, respectively. The water treatment at W60 negatively affected the plant growth and yield, resulting in reduced plant height, leaf number, leaf length and width, and a minor yield. Instead, fruit quality showed higher values of total soluble solids and titratable acidity. Conversely, plants watered with W80 showed results similar to the control (W100) in terms of development and yield. In conclusion, it is possible to assume that a reduction of water is desirable, guaranteeing economic and environmental gains for farmers.

**Keywords:** *Fragaria* × *ananassa*; irrigation; water stress; yield



## 1. Introduction

The urban population is going to increase by 2–3 billion by 2050 [1]. This demographic increase has an impact on the agricultural sector, caused by food demand. The human activity impact has determined an imbalance between resource exploitation and the natural regeneration capacity [2]. The use of products and services, without damaging the resources they come from, is the key to sustainability [3]. The food system is currently in a situation of globalization, depletion of resources, and climate change, which break the already-fragile natural balance. The major challenge for agriculture is to increase the quantity and the quality of products while reducing inputs (water and fertilizers) to guarantee environmental sustainability. The interest in healthy food is leading to a greater demand for natural products, including berries [4–6]. Strawberries play an important role in the diet, due to their quality and nutritional characteristics [7]. Strawberries are widely consumed, either fresh or used in the preparation of transformed foods. Fresh consumption is appropriate to ensure the maximum availability of nutrients, vitamins, and fibers. Strawberry fruits have biological properties and are beneficial for human wellness [8–13]. Some of the reasons farmers are cultivating strawberries are given by their positive characteristics, such as yield, resistance to biotic and abiotic stress, fruit quality, and the relevant profit [14,15]. The main factor of green sustainability is given by the knowledge of the plant's fertilization and water requirements. A balanced correlation between nutrients and the proper use of water

is essential for satisfactory plant growth. Consequently, the proper amount of input for efficient crop growth avoids the loss of nutrients and water in the environment.

Regarding the role of water in plant growth stages, this element affects the physiological and biochemical processes in all plant organs. A deficit of water intake leads to a decrease in chlorophyl content and consequently stunted growth and malformation of reproductive organs [16], combined with small fruits, that determine reduced yield [17–20]. The main percentage of strawberry fruit is constituted by water [21], and deterioration in the fruit's sensory qualities is determined by excessive water application [19–22]. In a dry climate, frequent and not-well-managed irrigation could lead to a negative impact on the fruit aroma [23]. Consequently, a proper irrigation system is a prerequisite for the attainment of a high yield of high-quality fruits. This is particularly true in protected systems, where artificial watering is the only source of water.

Therefore, strawberry cultivation systems need well-defined management of water application, an increasingly important resource in the near future.

Based on these considerations, this study aims to investigate the effect of different water regimes ($1120 \ \text{m}^3 \ \text{ha}^{-1}$, $891 \ \text{m}^3 \ \text{ha}^{-1}$, and $666 \ \text{m}^3 \ \text{ha}^{-1}$) on the growth, productivity, and qualitative responses of 'Cristina', 'Romina', and 'Sibilla' strawberry cultivars grown in an open field with the standard early spring cultivation cycle in Italy.

## 2. Materials and Methods

### 2.1. Plant Materials

Cold-stored plants of single-cropping cultivars ('Romina', 'Sibilla', and 'Cristina'), category A, were used for this trial. The growth material was obtained from an Italian nursery company, Coviro Soc. Cons. a.r.l. (Cervia, Italy). 'Romina' (early-ripening cultivar) and 'Cristina' (late-ripening cultivar) originated from the breeding program of UNIVPM D3A, while 'Sibilla' (medium-ripening cultivar) was provided by Consorzio Italiano Vivaisti (CIV). These cultivars show strong adaptability in different cultivation systems (open field and protected) and several areas of Italy [24]. 'Cristina' stands out for its very high productivity, and the fruits are large with a good taste [24]. 'Romina' is mainly valued for its sweet taste, correlated to the high sugar and low acidity. Many studies [10,24] confirm that the fruits of 'Romina' and 'Cristina' show high nutritional quality determined by high polyphenol, anthocyanin, vitamin C, and folate contents. 'Sibilla' is a cultivar suitable for European continental climates. The main positive aspects of 'Sibilla' are the high production potential with bright-red-colored fruit and a high level of firmness.

### 2.2. Site, Experimental Design and Treatments

The experimental trials were set up at the experimental farm of Assam (Agenzia per i Servizi nel Settore Agroalimentare della Regione Marche) located in Petritoli (FM), Marche Region, Italy (43°04′01.56″ N; 13°41′19.22″ E), for two consecutive cultivation cycles: 2016/2017 and 2017/2018. The planting was carried out in open-field conditions at the end of July in 2016 (for the first cycle) and in 2017 (for the second cycle) and covered with a plastic tunnel from mid-February until the end of June. The planting density was $5.5$ plants $\text{m}^{-2}$ in twin rows. The distance between and along rows was 30 and 35 cm, respectively. The non-fumigated soil was composed of 30% clay, 30% sand, and 40% silt at a pH value of 8.14 (Table 1). The fertigation was controlled with a Dosatron® D8R (Dosatron SAS, Tresses, FR). The irrigation system for each line was composed of two dripline hoses Toro® Acqua-Traxx FlowControl (model EAFCXxx0867) with emitter spacing of 20 cm, a flow rate at 0.7 bar of $5.07 \ \text{L} \ \text{h}^{-1} \ \text{m}^{-1}$, covered by plastic film. The cultivation was based on the Standard Integrated Pest Management (IPM) (Directive128/2009).

The fertilization plan was obtained with 10–52–10 (N–P–K) from August until March and 20–20–20 (N–P–K) from April to June for both the cultivation seasons. The scheduling irrigation was based on soil water potential. The irrigation for W100 started at $-20 \ \text{kPa}$ [25–27], for W80 at $-30 \ \text{kPa}$, and for W60 at $-40 \ \text{kPa}$. This parameter was recorded by six tensiometers, two for each irrigation regime. These instruments were

positioned between two plants in a row, at a soil depth of 15 cm, the root expansion zone. The probes used were Watermark® and were connected to a datalogger that recorded the measurements daily. The experiment consisted of a water reduction trial applied under three irrigation regimes (W). The control treatments (W100) corresponded to the recommended rates (Delibera 786-10 July 2017 of Marche Region) in the cultivation area; the other two regimes were 20% less (W80) and 40% less (W60) with respect to the recommended amounts of water (the total amount of water applied during both the seasons is shown in Table 2). The soil water potential was continuously monitored from the second week of March, at the flower initiation stage (stage 5 of the BBCH scale), to the second week of June, at the last harvest date (stage 8 of the BBCH scale), which correspond to the period of experimental inputs reduction. The measurement of this parameter helped to minimize the possible negative effect on plants' morphological and physiological development induced by reduced water restitution [28]. For each trial, the experimental design was a split-plot design, with 3 different treatments (main plots) and 3 cultivars (sub plots) for each year of study. Each sub-plot of a single cultivar contained 8 plants × 3 replicates. Each treatment comprised a total of 72 plants (Figure 1).

**Table 1.** Soil characteristics of the experimental field (Assam laboratory). Legend: U.M., unit of measurement.

| Trial Field | U.M. | Results | Method |
|---|---|---|---|
| pH | | 8.14 | D.M. 13/09/99 GU SO n.248 del 21/10/1999 III.1 |
| Sand | g Kg$^{-1}$ | 304 | D.M. 13/09/99 GU SO n.248 del 21/10/1999 II.5 |
| Silt | g Kg$^{-1}$ | 399 | D.M. 13/09/99 GU SO n.248 del 21/10/1999 II.5 |
| Clay | g Kg$^{-1}$ | 297 | D.M. 13/09/99 GU SO n.248 del 21/10/1999 II.5 |
| Active Limestone | g Kg$^{-1}$ | 61 | D.M. 13/09/99 GU SO n.248 del 21/10/1999 V.2 |
| Total Limestone | g Kg$^{-1}$ | 174 | D.M. 13/09/99 GU SO n.248 del 21/10/1999 V.1 |
| Assimilable P | g Kg$^{-1}$ | 3.7 | D.M. 13/09/99 GU SO n.248 del 21/10/1999 XV.3 |
| Exchangeable Na | g Kg$^{-1}$ | 15 | D.M. 13/09/99 GU SO n.248 del 21/10/1999 III.2, XIII.2.6 |
| Exchangeable Ca | g Kg$^{-1}$ | 4597 | D.M. 13/09/99 GU SO n.248 del 21/10/1999 III.2, XIII.2.6 |
| Cation exchange capacity | meq/100 g | 21.9 | D.M. 13/09/99 GU SO n.248 del 21/10/1999 III.2 |
| Assimilable iron | g Kg$^{-1}$ | 9.7 | D.M. 11/05/92 GU n.121 del 25/05/1992 Method n.37 |
| Assimilable Mn | g Kg$^{-1}$ | 4.1 | D.M. 11/05/92 GU n.121 del 25/05/1992 Method n.37 |
| Assimilable Zn | g Kg$^{-1}$ | 0.52 | D.M. 13/09/99 GU n.248 del 21/10/1999 XII.1 |
| Assimilable Cu | g Kg$^{-1}$ | 2.7 | D.M. 13/09/99 GU n.248 del 21/10/1999 XII.1 |
| Boron soluble | | 0.1 | D.M. 13/09/99 GU n.248 del 21/10/1999 XII.1 |
| C/N | | 7.7 | |
| Organic Matter | g Kg$^{-1}$ | 11.9 | D.M. 13/09/99 GU SO n.248 del 21/10/1999-VII.3. VII.3.6 |
| Total N | g Kg$^{-1}$ | 0.90 | D.M. 13/09/99 GU SO n.248 del 21/10/1999-XIV.2 + XIV.3 mod D.M. 25/03/2002 GU n.84 del 10704/2002 |
| Mg/K | | 2.7 | |
| Exchangeable Mn | mg Kg$^{-1}$ | 155 | D.M. 13/09/99 GU SO n.248 del 21/10/1999 XIII.2, XIII.2.6 |
| Exchangeable K | mg Kg$^{-1}$ | 410 | D.M. 13/09/99 GU SO n.248 del 21/10/1999 XIII.2, XIII.2.6 |

**Table 2.** Amount of water restitution during each cultivation cycle for the three treatments: W100 = 100% of water restitution; W80 = 80% of water restitution with respect to the standard (W100); W60 = 60% of water restitution with respect to the standard (W100).

| Irrigation Treatment | First Season (m$^3$ ha$^{-1}$) | Second Season (m$^3$ ha$^{-1}$) |
|---|---|---|
| W100 | 1104 | 1135 |
| W80 | 899 | 883 |
| W60 | 700 | 631 |

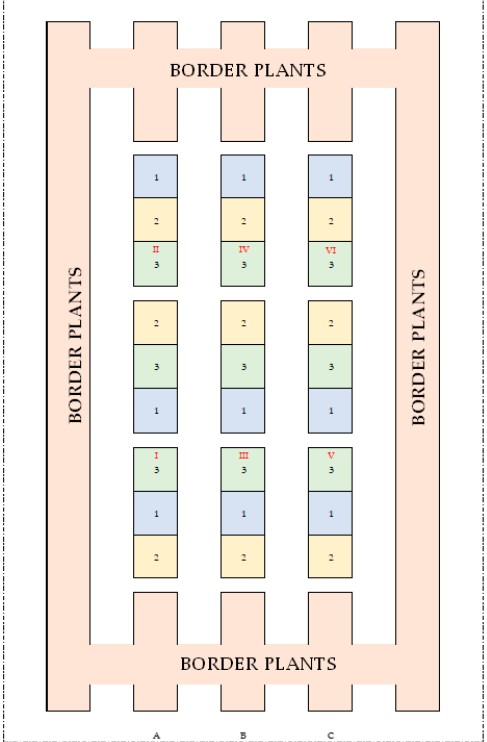

| N° Main plot | N° SubPlot | N° plants/plot | Total plants |
|---|---|---|---|
| A (100%) | 3(replica)×3(cultivar) | 8 | 72 |
| B (80%) | 3(replica)×3(cultivar) | 8 | 72 |
| C (60%) | 3(replica)×3(cultivar) | 8 | 72 |

**Legend**:

- Each number represents the studied cultivar: 1 'Romina', 2 'Sibilla', 3 'Cristina'
- Each number represents the tensiometer: I, II, III, IV, V, VI
- Each letter represents the water treatment: A=100%, B=80%, C=60%
- A=W100, B=W80, C=W60

**Figure 1.** Experimental design.

*2.3. Meteorological Data*

During the two years of the trial, meteorological data were detected, in particular, rainfall (Figure 2), solar radiation (Figure 3), and air temperature (Table 3).

(**a**)

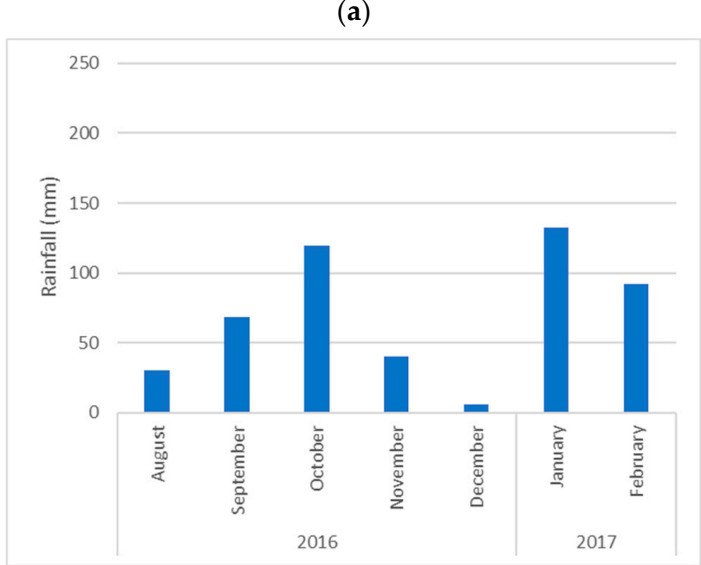

**Figure 2.** *Cont*.

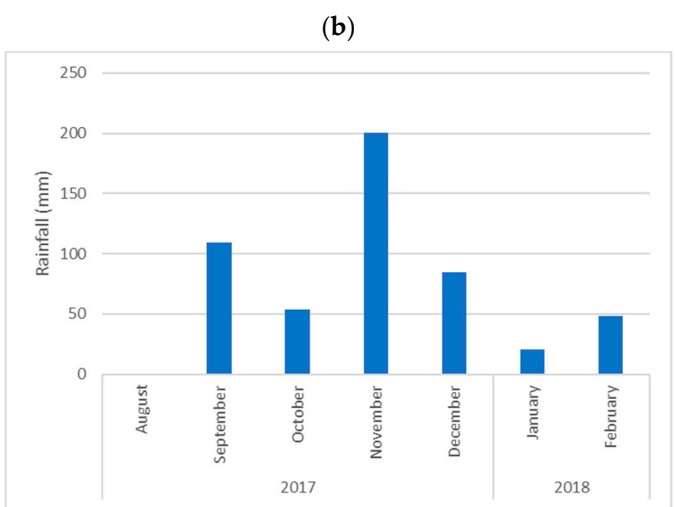

**Figure 2.** (**a**) Rainfall data of 1-year test: 264.2 mm of rain from August 2016 to December 2016; 213.8 mm from January 2017 to February 2017. (**b**) Rainfall data of 2-year test: 448 mm of rain from August 2017 to December 2017; 69 mm from January 2018 to February 2018.

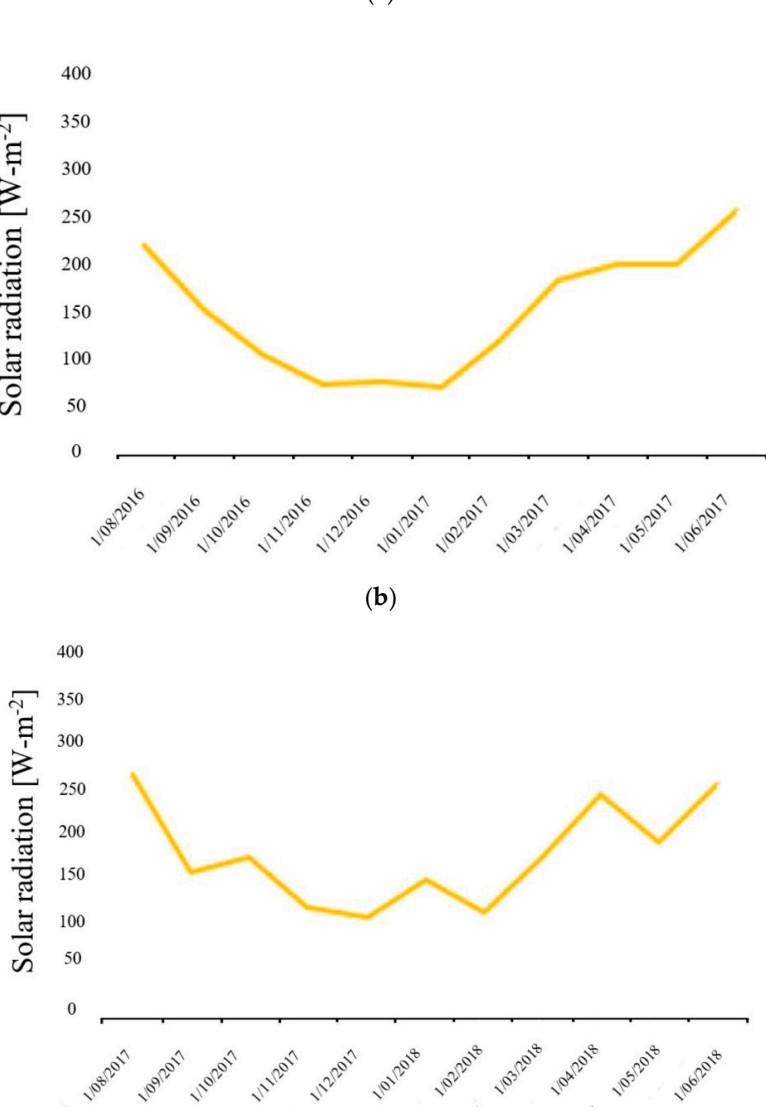

**Figure 3.** (**a**) Solar radiation 2016–2017. (**b**) Solar radiation 2017–2018.

**Table 3.** Air temperature data, 2017–2018.

| Year | Month | Medium Temperature (°C) | Maximum Temperature (°C) | Minimum Temperature (°C) |
|---|---|---|---|---|
| 2017 | March | 12.2 | 27.6 | 0.2 |
| | April | 14.6 | 31.4 | 0.1 |
| | May | 19.8 | 36.1 | 4.1 |
| | June | 25.6 | 42 | 10.9 |
| 2018 | March | 8.5 | 20.1 | 0 |
| | April | 15.9 | 26.6 | 7 |
| | May | 18.6 | 29.6 | 9.9 |
| | June | 22 | 31 | 14 |

*2.4. Analyzed Parameters*

2.4.1. Growth Parameters

The vegetative parameters were measured in triplicate on 8 plants of each plot for each cultivar, for both reduction trials. The parameters were analyzed once per cultivation cycle on 27 April 2017 and 30 April 2018. The values obtained in two different years were averaged. Branch crowns, inflorescences, and leaves were counted. Leaf size (length and width of the median leaf) and plant height were measured manually with the scale (ruler), expressed in cm.

2.4.2. Productive Parameters

The harvest dates of the first season were reported from 18th of April to 6th of June 2017, while the second season started on 30th of April and finished on 12th of June 2018 (Table 4). The Precocity Index (IP), Average Fruit Weight (AFW), total yield, and marketable production (fruits not smaller than Ø < 22 mm, not rotted, or deformed) were determined on average throughout the two harvest seasons. The methods used were described in Capocasa et al. [24].

**Table 4.** Harvesting dates for 2016/2017 and 2017/2018 seasons.

| Year | Month (Harvest Time) | Date |
|---|---|---|
| Season 2016/2017 | April | 18–21–26–28 |
| | May | 2–4–8–10–11–12–15–17–19–22–25–26–29 |
| | June | 1–6 |
| Season 2017/2018 | April | 30 |
| | May | 3–7–9–11–14–16–18–21–23–25–28–30 |
| | June | 1–4–6–8–12 |

2.4.3. Qualitative Parameters

Fruit color, firmness, total soluble solids, and titratable acidity were analyzed on ripe fruits of the first, second, and third main pickings. For each genotype/treatment thesis, ten fruits were selected at each harvest to evaluate the saturation of the color (Chroma) and the firmness with a Minolta Chromameter CR 400 (Konica Minolta, Tokyo, Japan) and a penetrometer 327 (Effegì, Ravenna, Italy), respectively. The CR-400 was used to evaluate the skin color of fresh fruits, measuring two points on opposite sides of each fruit, using CIELAB values (L*, a*, b). The Chroma was evaluated from a and b values $[(a^2 + b^2)]^{\frac{1}{2}}$. The Chroma measures the saturation of the color: A higher Chroma value represents pale fruits and low Chroma represents dark fruits. The different values of Chroma depend on the cultivar and stage of ripeness. The flesh firmness was measured on the same fresh fruits through the perforation of each fruit on two points of opposite faces, using a 6 mm star-tip probe. Data were expressed as g cm$^{-2}$. Fruits analyzed for color and firmness were subsequently stored at −20 °C for total soluble solids (TSS) and acidity titration analyses.

The frozen samples were then defrosted and hand-squeezed, obtaining strawberry juice. The (TSS), expressed in °Brix, was analyzed through a digital refractometer (PR-101α ATAGO, Tokyo, Japan) at 25 °C, placing one droplet of juice sample in the refractometer's measuring cell. The titratable acidity (TA) was measured through titration, which was performed on 10 g of strawberry juice previously obtained, added to 10 g of distilled water, with the addition of a pH indicator (bromothymol blue). The prepared sample was titrated with a 0.1 N NaOH solution, until pH 8.2, and the acidity was expressed as mEQ of NaOH per 100 g of Fresh Weight (FW).

### 2.5. Statistical Analyses

The data presented in this study regarding the strawberry plant and fruit parameters were presented as the average ± standard deviation (SD). A three-way analysis of variance was used to test the differences among the year of cultivation, cultivar, treatments, and corresponding interactions. Statistically significant differences in means were determined with the Fisher test (Least Significant Difference, LSD) ($p \leq 0.05$). Statistical processing was carried out using STATISTICA software (Stasoft, Tulsa, OK, USA).

## 3. Results and Discussion

### 3.1. Growth Parameters of Single-Cropping Cultivars with Reduced Water Supply

Assessing the growth parameters resulting from the different water treatments (Table 5), it appears that the year (a) exerted an important influence on the branch crown number, plant height, leaf number, and leaf width. The cultivar (b) was effective in the determination of the branch crown number, inflorescences number, plant height, leaf height, and leaf width. The interaction (a) × (b) swayed the branch crown number, inflorescences number, plant height, leaf number, leaf height, and leaf width. The interaction (a) × (c) was effective only for the leaf number. The interaction between (a) × (b) × (c) played an important role in the branch crown number, leaf number, and leaf width.

**Table 5.** Three-way analysis of variance (ANOVA) for the growth parameters ** = significant interaction with $p < 0.01$; * = significant interaction with $p < 0.05$; NS = not significant interaction.

| Parameter | Number of Crowns | Number of Inflorescences | Plant Height | Number of Leaves | Leaf Length | Leaf Width |
|---|---|---|---|---|---|---|
| Year (a) | ** | NS | ** | ** | NS | ** |
| Cultivar (b) | ** | ** | ** | NS | ** | ** |
| Treatment (c) | NS | NS | ** | ** | ** | ** |
| Year × Treatment (a) × (c) | NS | NS | NS | * | NS | NS |
| Cultivar × Treatment (b) × (c) | * | NS | NS | NS | NS | NS |
| Year × Cv × Treatment (a) × (b) × (c) | NS | NS | * | ** | NS | ** |

Generally, a reduction in water intake compromised the plant height, leaf number, leaf height, and leaf width. In fact, as described by Klamkowski and Treder [29], water reduction could negatively affect the above-ground part of the plant. Water allows the distribution of photo-assimilates, bringing a reduction in vegetative growth and a delay in reproductive organ development. The plant transfers the photosynthesized products to increase the number, length, volume, and dry weight of roots, allowing the plant to survive [30]. In our study, the W60 treatment caused an approximately 8% reduction in plant height and leaf number, compared to W100 (Table 6). Both leaf length and leaf width were influenced by water restriction (Table 7). In fact, reducing the water administration to 40% brought a decrease in leaf size. These results are in line with Yuan et al. [19] and Grant et al. [31], who described that leaf number increased proportionally to the amount of water administered. These results demonstrated the importance of an appropriate estimation of the water requirement, in order to restore a suitable amount of this source, for the beneficial effects of irrigation on strawberry plant growth [32]. The genotype appeared to affect the

plant height and leaf size. In fact, in all the treatments, 'Romina' and 'Sibilla' showed the tallest plants with the largest leaves, followed by the 'Cristina' cultivar. The water regime had no effect on plant branch crowns and inflorescences, likely due to the starting period of the differential irrigation after the flower initiation stage. A statistical difference was observed only with 'Cristina', which exhibited a statistically lower number of branch crowns than 'Romina' (Table 8), while 'Sibilla' had a statistically lower number of floral axes than 'Cristina' in W80. These results evidenced that the impact of water stress on plant vegetative development is strictly related to the genotype.

**Table 6.** Effects of water availability on plant height and leaf number in different strawberry cultivars. Values with the same lowercase letter for the same parameter were not statistically different for Fisher's LSD test ($p < 0.05$). Values are expressed as means of two years (2017–2018) ± standard deviation (SD).

| Cultivar | Plant Height | | | Number of Leaves | | |
|---|---|---|---|---|---|---|
| Treatment | W100 | W80 | W60 | W100 | W80 | W60 |
| 'Cristina' | 32.6 ± 4.4 d | 31.4 ± 4.3 d | 28.1 ± 3.6 e | 25.5 ± 7.2 ab | 24.6 ± 5.3 abc | 23.5 ± 7.1 bc |
| 'Romina' | 39.1 ± 3.1 a | 39 ± 3.0 a | 37 ± 3.1 bc | 25.1 ± 5.9 ab | 25.4 ± 5.3 ab | 22.5 ± 4.8 c |
| 'Sibilla' | 39.1 ± 5.1 a | 38.2 ± 6.3 ab | 36.2 ± 6.5 c | 26.4 ± 5.0 a | 24.7 ± 6.1 abc | 24.7 ± 5.4 abc |

**Table 7.** Effects of water availability on development of leaves in different strawberry cultivars. Values with the same lowercase letter for the same parameter were not statistically different for Fisher's LSD test ($p < 0.05$). Values are expressed as means of two years (2017–2018) ± standard deviation (SD).

| Cultivar | Leaf Length (cm) | | | Leaf Width (cm) | | |
|---|---|---|---|---|---|---|
| Treatment | W100 | W80 | W60 | W100 | W80 | W60 |
| 'Cristina' | 8.9 ± 0.7 d | 9 ± 0.9 de | 8.4 ± 0.9 e | 7.9 ± 0.9 c | 7.8 ± 0.9 c | 7.2 ± 1.0 d |
| 'Romina' | 9.5 ± 1.1 abc | 9.2 ± 1.0 bcde | 9.2 ± 1.1 cde | 9.4 ± 1.0 a | 9.4 ± 1.1 a | 9.2 ± 1.2 a |
| 'Sibilla' | 9.6 ± 1.1 ab | 9.8 ± 1.2 a | 9.3 ± 1.0 bcd | 8.4 ± 0.9 b | 8.1 ± 1.2 bc | 8.0 ± 0.8 c |

**Table 8.** Effects of water availability on branch crowns number and inflorescences number in different strawberry cultivars. Values with the same lowercase letter for the same parameter were not statistically different for Fisher's LSD test ($p < 0.05$). Values are expressed as means of two years (2017–2018) ± standard deviation (SD).

| Cultivar | Number of Crowns | | | Number of Inflorescences | | |
|---|---|---|---|---|---|---|
| Treatment | W100 | W80 | W60 | W100 | W80 | W60 |
| 'Cristina' | 4.0 ± 1.6 bc | 4.3 ± 1.4 abc | 3.9 ± 1.3 c | 12.9 ± 3.7 ab | 13.3 ± 3.3 a | 12.0 ± 4.3 abc |
| 'Romina' | 4.9 ± 2.3 a | 4.8 ± 2.0 ab | 5.0 ± 2.3 a | 12.0 ± 3.5 abc | 12.6 ± 2.9 abc | 12.0 ± 3.7 abc |
| 'Sibilla' | 4.9 ± 2.0 a | 4.2 ± 1.9 abc | 4.4 ± 1.9 abc | 11.4 ± 4 bc | 11.4 ± 4.1 c | 11.2 ± 3.7 c |

*3.2. Productive Parameters of Single-Cropping Cultivars with Reduced Water Supply*

Evaluating the productive parameters (Table 9), it stands out that the year (a), cultivar (b), and treatment (c), as well as the interaction of (a) × (b), were crucial for the precocity index, average fruit weight (AFW), and marketable and total production determination. The relations (a) × (c) and (b) × (c) did not highlight significant effects for any parameter. The interaction (a) × (b) × (c) only impacted AFW.

**Table 9.** Three-way analysis of variance (ANOVA) for the productive parameters ** = significant interaction with $p < 0.01$; * = significant interaction with $p < 0.05$; NS = not significant interaction.

| Parameter | Precocity Index | Average Fruit Weight | Marketable Production | Total Yield |
|---|---|---|---|---|
| Year (a) | ** | ** | ** | ** |
| Cv (b) | ** | ** | ** | ** |
| Treatment (c) | ** | * | ** | ** |
| Year × Cv (a) × (b) | ** | ** | * | ** |
| Year × Treatment (a) × (c) | NS | NS | NS | NS |
| Cv × Treatment (b) × (c) | NS | NS | NS | NS |
| Year × Cv × Treatment (a) × (b) × (c) | NS | ** | NS | NS |

Both the precocity index and AFW did not show significant differences among different water supply trials. In optimal conditions of water supply (W100), 'Romina' proved to be an early cultivar (IP = 129), 'Sibilla' an intermediate cultivar (IP = 134), and 'Cristina' a very late cultivar (IP = 146) (Table 10). These ripening periods denoted important statistical differences among cultivars, and those differences were maintained among the three treatments. For all the cultivars, a higher water reduction (W60) caused early ripening of the fruit (2 days in 'Sibilla' and approximately 1 day in 'Cristina' and 'Romina'), but the effect was not significant. More specifically, some studies demonstrated the active role of the planting [33] and cultivation [34] system on the ripening time of fruits. Among the tested cultivars, 'Cristina' had the statistically highest average fruit weight under all irrigation treatments (Table 10). Comparing W100 with the trials at a reduced water supply, similar trends were detected for all the cultivars. Furthermore, some other studies showed that berry size was not affected by the period of water shortage [35]. The study conducted by Ariza et al. [36] showed that the fruit weight did not change in six different strawberry cultivars placed in a tunnel, under different deficit irrigation. In our case, the fruit weight slightly decreased only in 'Romina' and 'Sibilla', although these changes were not significant. These results were in contrast with some studies, which showed a fruit weight decrease, reducing the administrated water [31,33,37]. This is presumably because the water flowing through the plant vascular system, in drought conditions, minimizes biophysical, metabolic, and hormonal factors entailed in cell turgor, osmotic pressure, and cell-wall extension [35]. The reduction in marketable production and total yield per plant is proportional to the decrease in the administered water, with a significant difference in the studied cultivars (Table 11). Similar results were obtained in the experiment conducted by Martinez-Ferri et al. [38], which highlighted significant losses in strawberry yields resulting from a water shortage of 30% of the plant's water requirement. In our study, the total and marketable weight loss showed statistical relevance between W100 and W60 treatments, with a decrease of 24% in 'Cristina', 19% in 'Romina', and 17% in 'Sibilla' (Table 11). However, the marketable production was strongly related to the genotype effect. The same trend was registered in each cultivar for the total production. The loss of production at W60 was 19% in 'Cristina', 17% in 'Romina', and 13% in 'Sibilla', with respect to W100. However, lowering the administrated water by 20% did not bring statistical differences among the studied cultivars regarding both marketable and total yield. Interestingly, the study conducted by Mezzetti et al. [39] highlighted an increase of 32% in strawberry yield correlated to the reduction of approximately 28% of the water supply commonly applied by an agricultural company in the South of Spain. Another aspect faced by Tunc et al. [40] is the importance of the accurate selection of the proper genetic material derived by the breeding selection, with high adaptability to drought conditions. In addition, the plant type used plays an important role in terms of vegetative development and adaptability to extreme conditions. On the other hand, as shown by Tunc et al. [41], an excessive water reduction in semi-arid conditions caused a reduction in strawberry yield. The genotype remained the main factor controlling plant yield capacity under water stress conditions.

'Cristina' showed greater marketable production at each treatment, resulting in being statistically higher than 'Romina' and 'Sibilla', which, in turn, were similar to each other. Furthermore, 'Cristina' appeared to be the most susceptible to a greater water reduction amount. In fact, the significant difference in overall total production at W100 between 'Cristina' and 'Romina' became not significant at W60. Overall, a reduction in irrigation caused a marked reduction in fruit production in the cultivar with high yield potential. This tendency could be attributed to the decrease in the plant crop water use efficiency under water shortage [38].

**Table 10.** Effects of water availability on precocity index and average fruit weight in different strawberry cultivars. Values with the same lowercase letter for the same parameter were not statistically different for Fisher's LSD test ($p < 0.05$). Values are expressed as means of two years (2017–2018) ± standard deviation (SD).

| Cultivar | Precocity Index (Days) | | | Average Fruit Weight (g) | | |
|---|---|---|---|---|---|---|
| Treatment | W100 | W80 | W60 | W100 | W80 | W60 |
| 'Cristina' | 146.3 ± 2.6 a | 145.6 ± 2.8 a | 145.2 ± 2.7 a | 30.2 ± 2.0 a | 30.2 ± 1.7 a | 29.8 ± 3.4 a |
| 'Romina' | 129.6 ± 3.1 cde | 128.1 ± 3.4 e | 128.2 ± 3 de | 17.9 ± 1.7cd | 16.6 ± 1.5 d | 16.5 ± 0.8 d |
| 'Sibilla' | 134.0 ± 3.6 b | 132.5 ± 4.4 bc | 132.1 ± 4.2 bcd | 20.0 ± 1.3 b | 20.1 ± 0.5 b | 19.0 ± 1.4 bc |

**Table 11.** Effects of water availability on marketable production and total production in different strawberry cultivars. Values with the same lowercase letter for the same parameter were not statistically different for Fisher's LSD test ($p < 0.05$). Values are expressed as means of two years (2017–2018) ± standard deviation (SD).

| Cultivar | Marketable Production (g plant$^{-1}$) | | | Total Yield (g plant$^{-1}$) | | |
|---|---|---|---|---|---|---|
| Treatment | W100 | W80 | W60 | W100 | W80 | W60 |
| 'Cristina' | 755.1 ± 110.1 a | 694.0 ± 143 a | 573.8 ± 151.8 b | 856.2 ± 121.2 a | 809.9 ± 138.6 ab | 689.6 ± 162.9 cd |
| 'Romina' | 505.6 ± 53.3bcd | 494.3 ± 66.6 bcd | 409.8 ± 51.1 d | 695 ± 51.4 c | 712.0 ± 75.9 bc | 578.6 ± 45 d |
| 'Sibilla' | 555.6 ± 54.9 bc | 465.5 ± 69.8 cd | 461.1 ± 36 cd | 809.8 ± 76.9 ab | 707.4 ± 78.1 bc | 703.9 ± 57.5 bc |

### 3.3. Qualitative Parameters of Single-Cropping Cultivars with Reduced Water Supply

Evaluating the fruit qualitative parameters (Table 12), the role of the year (a), cultivar (b), treatment (c), and the interaction between (a) × (b) was evident in relation to sugar content, titratable acidity, firmness, brightness L*, redness a*, yellowness b*, and the chroma index. The influence of (a) on chroma and (c) on firmness is not significant. The relation between (a) × (c) and (b) × (c) had no significant effect on the studied fruit quality parameters. The interaction between (a) × (b) × (c) had no influence on any of the evaluated parameters, except for the fruit soluble sugar content.

The genotype had a great influence on the soluble sugar content of the fruits [41]. In fact, independently of the provided treatment, 'Sibilla' fruits showed the highest soluble solids content, followed by 'Romina', and lastly, 'Cristina'. A reduction in the water supply amount corresponded to an increase in fruit soluble solid content, which was significantly different between W100 and W60. Fruits of 'Sibilla', 'Romina', and 'Cristina' displayed an increase of 0.5°Brix at W60 compared to W100. This result pointed out the effect of water stress on fruit sugar content in all genotypes. The results presented here were supported by other studies [28,37] that demonstrated the correlation between the water stress effect and the rise in sugar content, considered an important parameter by consumers. However, another study by Saied et al. [42] differed from our findings, resulting in a solid soluble content decrease under deficit irrigation conditions. Ripoll et al. [35] described a difference between fresh and dry fruit, considering the correlation between sugars and acids. In a drought environment, fresh strawberries increased the sugar content and maintained acids, and instead, the dry fruit remained unchanged in terms of sugars and decreased

acidity value. The titratable acidity is a fruit quality descriptor, principally depending on the genotype. 'Cristina' and 'Romina' fruits showed less acidity than 'Sibilla' in W100 and W80 (Table 13). Evaluating the 'Cristina' cultivar, the decrease in water input brought an increase in fruit acidity content, whereas 'Romina' and 'Sibilla' remained stable in all the water regimes. The results of these two last cultivars were in contrast with the results found in other studies [28,33], which demonstrated that water stress caused a decrease in fruit acidity. According to Bordonaba and Terry [28], the deficit of the irrigation treatment could cause a different genotype-dependent effect on the acid's metabolism. In fact, different respiratory metabolisms among cultivars entailed different utilization of substrates such as acids. 'Sibilla' stood out for its utmost firmness, followed, respectively, by 'Romina' and 'Cristina', in all the irrigation regimes (Table 14). However, the water stress seemed not to influence the fruit's firmness, which is an important factor for consumer acceptance and the fruit's shelf-life. These results are in opposition to another study [33], which described a correlation between the growing condition and fruit firmness. The fruits obtained in the stress condition showed a higher firmness compared to the control treatment. A possible theory is given by the correlation between irrigation and firmness: in fact, cell dimension, solute transportation, and accumulation through the cell are directly dependent on the water entering the plant [35]. Fruit color is an important factor for consumers' acceptance, dependent on the genotype. In the present study, 'Cristina' exhibited the lowest values of Chroma, L*, a*, and b*, in response to all the treatments (Tables 14 and 15, respectively), in comparison with 'Sibilla' and 'Romina'. The color of 'Cristina' and 'Romina' fruit was unaffected by the water treatment. These results were partially in line with Adak et al. [33], who revealed that fruit color values (L*, Chroma, a*) were not influenced by the water intake. These results have been partially confirmed by our study: The Chroma values of 'Romina' and 'Cristina' fruits did not change under water stress conditions in comparison to the control, but the fruits of 'Sibilla' showed significantly increased darkness in their color with water reductions of 20% both at W80 and W60 with respect to W100. The fruit of 'Sibilla' was the most influenced by water administration. In fact, it highlighted a significant decrease in L*, a*, b*, and Chroma at W80 and W60 compared to W100. Bordonaba and Terry [28] stated that Chroma values were lower in fruits subjected to water stress.

**Table 12.** Three-way analysis of variance (ANOVA) for the qualitative parameters ** = significant interaction with *p* < 0.01; * = significant interaction with *p* < 0.05; NS = not significant interaction.

| Parameter | Sugar Content | Titratable Acidity | Firmness | L* | a* | b* | Chroma |
|---|---|---|---|---|---|---|---|
| Year (a) | ** | ** | ** | ** | ** | ** | NS |
| Cv (b) | ** | ** | ** | ** | ** | ** | ** |
| Treatment (c) | ** | * | NS | * | * | ** | ** |
| Year × Cv (a) × (b) | ** | ** | ** | ** | ** | ** | ** |
| Year × Treatment (a) × (c) | NS | NS | NS | NS | NS | NS | NS |
| Cv × Treatment (b) × (c) | NS | NS | NS | NS | NS | NS | NS |
| Year × Cv × Treatment (a) × (b) × (c) | ** | NS | NS | NS | NS | NS | NS |

**Table 13.** Effects of water availability on sugar content and titratable acidity in different strawberry cultivars. Values with the same lowercase letter for the same parameter were not statistically different for Fisher's LSD test ($p < 0.05$). Values are expressed as means of two years (2017–2018) $\pm$ standard deviation (SD).

| Cultivar | Sugar Content (Brix°) | | | Titratable Acidity (meqNaOH 100 g$^{-1}$ Fruit Weight) | | |
|---|---|---|---|---|---|---|
| Treatment | W100 | W80 | W60 | W100 | W80 | W60 |
| 'Cristina' | 6.6 ± 0.4 f | 6.9 ± 0.6 f | 7.1 ± 0.8 ef | 10.4 ± 1.1 d | 10.9 ± 1.1 bcd | 11.3 ± 1.6 abc |
| 'Romina' | 7.6 ± 0.8 de | 7.7 ± 1 cd | 8.1 ± 1.1 bcd | 10.7 ± 0.7 cd | 10.7 ± 1.1 cd | 11.1 ± 1.0 abc |
| 'Sibilla' | 8.2 ± 0.9abc | 8.6 ± 1.2ab | 8.8 ± 1.3 a | 11.5 ± 1.0 ab | 11.7 ± 1.1 a | 11.7 ± 1.1 a |

**Table 14.** Effects of water availability on fruit firmness and chroma in different strawberry cultivars. Values with the same lowercase letter for the same parameter were not statistically different for Fisher's LSD test ($p < 0.05$). Values are expressed as means of two years (2017–2018) $\pm$ standard deviation (SD).

| Cultivar | Firmness (g/cm$^2$) | | | Chroma | | |
|---|---|---|---|---|---|---|
| Treatment | W100 | W100 | W80 | W60 | W80 | W60 |
| 'Cristina' | 271.7 ± 42.9 c | 45.1 ± 3.4 e | 44.7 ± 3.3 e | 44.2 ± 2.9 e | 286.1 ± 42.8 c | 275.5 ± 31.7 c |
| 'Romina' | 342.8 ± 38.6 b | 49.8 ± 1.4 cd | 48.5 ± 1.4 d | 49.5 ± 1.7 d | 355.2 ± 24.1 b | 367.2 ± 32.4 b |
| 'Sibilla' | 410.6 ± 84.0 a | 53.3 ± 1.5 a | 51.4 ± 2.2 b | 51 ± 2.0 bc | 409.3 ± 94.1 a | 415.5 ± 90.0 a |

**Table 15.** Effects of water availability on L*, a*, and b* in different strawberry cultivars. Values with the same lowercase letter for the same parameter were not statistically different for Fisher's LSD test ($p < 0.05$). Values are expressed as means of two years (2017–2018) $\pm$ standard deviation (SD).

| Cultivar | L* | | | a* | | | b* | | |
|---|---|---|---|---|---|---|---|---|---|
| Treatment | W100 | W80 | W60 | W100 | W80 | W60 | W100 | W80 | W60 |
| 'Cristina' | 38.1 ± 2.5 cde | 37.8 ± 2.2 de | 37.6 ± 2.2 e | 38.6 ± 2.2 d | 38.3 ± 2.1 d | 38.1 ± 1.9 d | 23.1 ± 3.1 d | 22.9 ± 3.2 d | 22.1 ± 2.9 d |
| 'Romina' | 39.5 ± 2.4 bc | 38.3 ± 2.4 cde | 39.2 ± 2.4 bcd | 41.2 ± 0.7 c | 40.5 ± 0.7 c | 41 ± 1.1 c | 27.9 ± 2.2 bc | 26.5 ± 2.1 c | 27.6 ± 1.9 bc |
| 'Sibilla' | 41.7 ± 2.6 a | 40.4 ± 2.5 ab | 40.0 ± 2.6 b | 43.6 ± 0.6 a | 42.6 ± 1.0 b | 42.5 ± 1.0 b | 30.6 ± 2.3 a | 28.6 ± 2.8 b | 28.2 ± 2.4 b |

## 4. Conclusions

The objective of this study was to verify any changes in growth, productivity, and qualitative responses of single-cropping strawberry cultivars with different water supplies and determine an acceptable trade-off between positive plant performance and a low environmental impact. Regarding water administration, among the tested cultivars, 'Sibilla' did not show significant variations for the vegetative parameters. 'Romina', followed by 'Cristina', was strongly influenced by the water reduction, in terms of leaf number and plant height. 'Cristina' displayed an evident leaf length and width reduction. The productive parameters decreased in response to higher water stress. The only exception is 'Sibilla', which presented similar yield values among different water trials. The qualitative parameters were also increasingly affected by water shortage in sugar and acidity, but not in firmness. In summary, it is possible to affirm that all three single-cropping studied cultivars, already adapted to the cultivation environment, maintained regular plant development, yield, and fruit quality at an 80% water supply. Upon further reduction (−40%), 'Romina' and 'Cristina' suffered a decrease in plant yield, while 'Sibilla' presented lower-quality fruits, in particular in terms of color (L*, a*, b* and Chroma).

In conclusion, if a company's orientation is to improve the quality characteristics of fruit, a strong water reduction could be evaluated, but this will negatively affect the plant yield. New breeding programs can be aimed to release new more resilient cultivars able to maintain good plant yield and fruit quality even at lower water restitution. Furthermore, from this study, the possibility of using tensiometers, which are cheap and easy to use, for

monitoring the soil water potential has emerged, as an indicator of the irrigation schedule. This information could be useful for the introduction of procedures that facilitate the rational management of water resources.

**Author Contributions:** Conceptualization: F.C. and B.M.; methodology: F.C. and M.M.; software: F.C., M.M. and D.R.; validation: F.C., L.M. and B.M.; formal analysis: M.M., F.B., V.P. and D.R.; investigation: F.C., M.M., L.M. and D.R.; resources: F.C. and B.M.; data curation: M.M., F.C. and D.R.; writing-original draft preparation: M.M. and D.R.; writing-review and editing: B.M. and L.M.; Mazzoni L.; visualization: M.M., F.C. and D.R.; supervision: F.C. and B.M.; project administration: F.C.; funding acquisition: B.M. and F.C. All authors have read and agreed to the published version of the manuscript.

**Funding:** This project received funding from the GOODBERRY project supported by the European Union's Horizon 2020 research and innovation program under grant agreement No. 679303.

**Conflicts of Interest:** The authors declare no conflict of interest.

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
