# Peer review of "Evaluation of Single-Cropping under Reduced Water Supply in Strawberry Cultivation"

_agronomy, doi:10.3390/agronomy12061396_

Round 1

Reviewer 1 Report

Comments:

Introduction: there is a lack of digital information on why they use these fertilizer doses, change the water level of productivity and the optimal amount of fertilizer.

There is a typographical error in this text. Standardizes the units of measure in text.

L. 4: the author doesn't write well. Franco capocassasa.

L140. Standardized metric unit (in the whole text), no standard or no standard?

L 162. Figure 2 is a table.

L 168. Check for errors in chemical formulas during writing.

L297。 Clarify why total production (e.g. "Christina") table 10, = 869.4 g / plant = saleable production (g / plant) + non saleable production (g / plant) = 783.1 + 101.1 is different from 869.4

L.236. Tables 5, 6, 7, 9, 10 and 11 showed that "there was no significant difference in the mean value of each parameter of Fisher LSD test (P < 0.05)", but there was no or no analysis.

L. 301。 Table 11: I think there is an error in the middle separator. Please check it.

Check the meaning of "N5" in the table. Query the average of N100, N80 and N60.

L. 543: conclusion: you must focus on drawing conclusions based on your results. The author provides more information. Improvements are needed because 75% of the texts correspond to published doctoral thesis paragraphs: marcellini et al., 2019. Genotype evaluation of sustainable utilization of strawberry cultivation resources University of technology in March.

L. 571: 10 vehicles detected.

Author Response

Please see the attachment "Response to reviewer 1".

Thank you.

Reviewer 2 Report

The article "Evaluation of single-cropping for sustainable resource use in strawberry cultivation" is current and well written. My recommendation would be to express more detailed requirements for nutrients (especially nitrogen) according to the BBCH scale.
Congratulations for your work!

Author Response

Please see the attachment "Response to reviewer 2".

Thank you.

Reviewer 3 Report

The authors demonstrated that reducing the input of water or nitrogen may not affect growth and fruit yield of strawberry. However, there are a lot of articles similar to the manuscript. Moreover, I did not find novelty in the treatments.

I have major comments for this manuscript.

  1. You must display climate data.
  2. You should make comparisons between treatments in the tables, not between cultivars.
  3. You should consider if all data is needed in this manuscript.
  4. You must use scientific words and units.
  5. You must review the references.

My minor comments are in the file.

Thanks,

Author Response

Please see the attachment "Response to reviewer 3".

Thank you.

Reviewer 4 Report

I have read the manuscript „Evaluation of single-cropping for sustainable resource use in strawberry cultivation” prepared by Micol Marcellini et al. for  Agronomy MDPI.

Although generally the experiments were good performed and the results have scientific value and importance, in my opinion the description of the applied methodology, and the way, in which results are presented and interpreted at the statistical level,  need major revision. In the present form the manuscript is unsuitable for publication. Some very important methodological informations are missing, and the interpretation of results seems to be very often incompatibile with the statistical analysis.

I am sending the pdf file with the manuscript, in which all my detailed comments and suggestions are written. The Authors should obligatory follow them to improve the manuscript.

Author Response

Please see the attachment "Response to reviewer 4".

Thank you.

Reviewer 5 Report

Agronomy 1699083
The above paper examines the effect of nitrogen and irrigation on the performance of three cultivars of strawberry over two years in Italy.

There is a major problem with the nitrogen study, with only three treatments, no data on plant nitrogen levels, and mostly no significant differences in nitrogen treatment means, pooled over the three cultivars. For instance, average marketable yields across the three nitrogen treatments are
similar (Table 10).

The results from the study are only applicable to the site used for the experiment, and cannot be readily transferred to other locations, in the absence of data on plant nitrogen concentrations. This is also made more difficult because the authors only used three treatments.

So, I suggest that the nitrogen study is not enough.

The work on irrigation is a little better, with significant differences in the general means, pooled across the three cultivars.

I am not sure what the authors should recommend as far as the best strategy to achieve the highest marketable yields. Overall, the response is not consistent across the three cultivars. Averaged across the three cultivars, marketable yields are similar in the W100 and W80 treatments and similar
in the W80 and W60 treatments. So, it is difficult to decide whether the W100 or W80 treatment is best.

There is another issue with the irrigation study is that no data are presented on plant water status, limiting the interpretation and universality of the results.

For some reason, the authors do not present data on rainfall or evapotranspiration, so it is not possible to determine the relative water defects in the three treatments.

There are too many tables. Some of the data sets should be combined.

The paper is too large to be readily digested.

The authors do not indicate a clear recommendation for nitrogen or irrigation management
(conclusions are very vague).

Many of the differences in plant growth and fruit quality amongst the treatments are small and probably not commercially significant, and probably reflect the experimental design (three-way factorials).

To summarise, the irrigation study is possible salvageable, but requires significant work by the authors.

Author Response

Please see the attachment "Response to reviewer 5".

Thank you.

Round 2

Reviewer 4 Report

The Authors did not revise and correct sufficiently the statistical interpretation, data presentation and further results description, as was suggested earlier. There are still mismatches concerning the significant/non-significant interactions between cultivar and treatments in general, and then "letters a,b, c/N.S" signed to mean values of particular traits presented in following tables (see detailed comments in attached file).

What are L*, a* and b* presented in tables 23 and 26?

Author Response

 "Please see the attachment for the comments, for the other modifications we are uploading the manuscript".

Reviewer 5 Report

Agronomy 1699083 R1

The authors have revised the above submission slightly.  There appears to be some hesitancy in taking on board the suggestions and comments by the reviewers.

I still believe that the nitrogen component should be removed from the paper.  The work itself is rather basic with numerous publications on the subject.  It doesn’t add significantly to the literature on the crop.  The study is handicapped by the lack of data on plant nitrogen status.  The results/recommendations for nitrogen applications are only applicable to the study area.  This dramatically reduces the impact of the study to scientists and agronomists in other growing areas.

No information has been provided on the water deficits experienced by the plants in the different treatments (rain + irrigation – evapotranspiration).  The authors have no data on plant water status as well.  Overall, the irrigation study is fairly superficial.

There is still too much text, too many tables and an excess of citations.

I suggest that the authors amend the paper using only the irrigation study, include data on the water deficits experienced by the plants, and reduce the size of the manuscript.   The revision should be sent out to new reviewers to see if it meets the minimum standard for the journal.

Author Response

 "Please see the attachment for the comments. For the modifications of the manuscript, we are uploading the manuscript".
